# A Novel One-Pot Synthesis of Poly(Propylene Carbonate) Containing Cross-Linked Networks by Copolymerization of Carbon Dioxide, Propylene Oxide, Maleic Anhydride, and Furfuryl Glycidyl Ether

**DOI:** 10.3390/polym11050881

**Published:** 2019-05-14

**Authors:** Lijun Gao, Xianggen Chen, Xiangjun Liang, Xiuzhi Guo, Xianling Huang, Caifen Chen, Xiaodan Wan, Ruyu Deng, Qifeng Wu, Lingyun Wang, Jiuying Feng

**Affiliations:** 1School of Chemistry and Chemical Engineering, Key Laboratory of Clean Energy Materials Chemistry of Guangdong Higher Education Institutes, Resource and Chemical Engineering Technology Research Center of Western Guangdong Province, Lingnan Normal University, Zhanjiang 524048, China; gljtalk@163.com (L.G.); cxgzgah@163.com (X.C.); xjliang1997@163.com (X.L.); guoxzls@163.com (X.G.); hangxl95@163.com (X.H.); caifenchen96@163.com (C.C.); xdwan1993@163.com (X.W.); ruyudeng@163.com (R.D.); qifenwu@163.com (Q.W.); 2Key Laboratory of Functional Molecular Engineering of Guangdong Province, School of Chemistry and Chemical Engineering, South China University of Technology, Guangzhou 510641, China; lingyun@scut.edu.cn

**Keywords:** poly(propylene carbonate), networks, copolymerization, modification

## Abstract

The thermoplastic poly(propylene carbonate) (PPC) containing cross-linked networks was one-pot synthesized by copolymerization of carbon dioxide, propylene oxide (PO), maleic anhydride (MA), and furfuryl glycidyl ether (FGE). The copolymers were characterized by Fourier transform infrared spectroscopy (FT-IR), gel permeation chromatography (GPC), differential scanning calorimetry (DSC), and thermogravimetric analysis (TGA) measurements. The thermal and dimensional stability of the copolymers were improved. When the MA and FGE load increased from 1 mol% to 4 mol% of PO, the copolymers contained the gel contents of 11.0%–26.1% and their yields were about double that of the PPC. The 5% weight-loss degradation temperatures (*T*_d,-5%_) and the maximum weight-loss degradation temperatures (*T*_d,max_) increased from 149.7–271.3 °C and from 282.6–288.6 °C, respectively, corresponding to 217.1 °C and 239.0 °C of PPC. Additionally, the hot-set elongation tests showed that the copolymers exhibited elasticity and dimensional stability with the minimum permanent deformation of 6.5% which was far less than that of PPC of 157.2%, while the tensile strengths were a little lower than that of PPC because of the following two conflicting factors, cross-links and flexibility of the units formed by the introduced third monomers, MA and FGE. In brief, we provide a novel method of one-pot synthesis of PPC containing cross-linked networks. According to this idea, the properties would be more extensively regulated by changing the cross-linkable monomers.

## 1. Introduction

Poly(propylene carbonate) (PPC) derived from carbon dioxide and propylene oxide (PO) has been drawing much attention in both academic and industrial fields [1,2] as a biodegradable polymer. However, PPC still has considerable limitations, for example, its low decomposition temperature, low glass transition temperature (*T*_g_), and its amorphous nature that severely limits its thermal stability and practical application [3,4]. It becomes brittle at low temperature and quickly loses strength at elevated temperature. For these reasons, significant efforts have been devoted to the modification of PPC. Crystallization of PPC was first expected to improve its properties, however it is very difficult to crystallize PPC even the stereoregular PPC was synthesized [5,6]. Regarding the thermal stability, end-capping with maleic anhydride, benzoyl chloride, ethyl silicate, acetic anhydride, and phosphorus oxychloride, or phosphoric ester can improve the decomposition temperature of PPC by converting PPC’s end-hydroxyl groups into other groups and suppressing the unzipping degradation of PPC [7,8]. In addition, terpolymerization with co-monomers [9,10,11,12], cross-linking [13,14,15] and fabrication with other polymers [16,17,18,19,20], inorganic fillers [21,22,23,24,25], or organic compounds [26,27,28,29] have been used to improve its thermal and mechanical performance. In all methods, cross-linked PPCs, whether physical or chemical, display good thermal stability and mechanical strength, especially dimensional stability at high temperature. These methods can effectively solve the cold-flow problem of PPC although the *T*_g_ is low. For example, a small amount of graphene oxide nanosheets (1 wt%) can form physical cross-links in a PPC matrix, which greatly enhances PPC [21]. The PPC chemically cross-linked with allyl glycidyl ether displayed excellent dimensional stability with the hot-set elongation at 65 °C of 17.2% and permanent deformation approaching zero compared with 35.3% and 17.2% for uncross-linked PPC, respectively [13]. Cross-linking customarily requires two steps, including introducing a cross-linkable moiety like double bonds into the PPC backbone and subsequently cross-linking using radical initiators. As an alternative, the reaction between PPC and isocyanate can produce partially cross-linked PPC [30]. It was also prepared by the electron-beam irradiation of blending of PPC and polyfunctional monomers such as trimethylopropane triacrylate, pentaerythritol triacrylate, and poly(ethylene glycol) dimethyl methacrylate [31]. Organic silylated PPC following hydrolysis reaction can also form cross-linked PPC [32,33]. Compared with the two-step method, PPC can be obtained directly by one-step terpolymerization of PO and CO_2_ with diepoxides [15,34], dianhydrides [35,36], or mixed monomers [37]. Sometimes, the addition of diepoxides in the CO_2_/PO copolymerization does not form cross-linked PPC [38,39], which may be related to the reactivity of the diepoxides.

Hilf and coworkers prepared cross-linked poly((furfuryl glycidyl ether)-co-(glycidyl methyl ether) carbonate) (P((FGE-*co*-GME)C)) copolymers using a two-step method [40]. They first prepared the P((FGE-*co*-GME)C) copolymers with furyl pandants by terpolymerization of CO_2_, FGE, and glycidyl methyl ether (GME). Cross-linking was conducted via a Diels–Alder (D–A) reaction between the furyl pandants of P((FGE-*co*-GME)C) and maleimide derivatives which were introduced in the second step. They focused on the synthesis of P((FGE-co-GME)C) and there was no mention of the thermal stability or mechanical properties of the copolymers. Inspired by their research, we realized the one-pot synthesis of PPC with cross-linked networks by the copolymerization of CO_2_, PO, MA, and FGE. Unlike diepoxides or dianhydrides reported previously, the additional monomers contain both epoxide/anhydride and other mutually reactive groups. The former participate in the CO_2_/PO copolymerization and introduce the reactive groups into PPC’s pendants. In this case, MA and FGE can introduce both the C=C double bonds and furan groups into the PPC’ chains. They react with each other and generate the networks during the copolymerization. The thermal and mechanical properties and dimensional stability of the obtained copolymers were fully investigated in this work.

## 2. Materials and Methods

### 2.1. Materials

PO was refluxed over calcium hydride for 8 h, distilled under dried nitrogen gas and stored over 0.4 nm molecular sieves prior to use. CO_2_ of 99.99% purity was commercially obtained without further purification. MA, glycerol, furfuryl alcohol (FA), tetrabutylammonium bromide (TBAB), glutaric acid, and zinc oxide were all purchased from Aladdin Industrial Corporation and directly used without further purification. FGE was synthesized according to the literature [41]. The ^1^H NMR spectrum is seen in Appendix A. Zinc glutarate (ZnGA) was synthesized according to the literature [42]. All other reagents and solvents were of analytical grade and used without further purification.

### 2.2. General Copolymerization Procedure

The copolymerization was conducted in a 100 mL autoclave reactor equipped with a magnetic stirrer. The 0.1 g zinc glutarate (ZnGA) catalyst and a certain proportion of MA and FGE were added to the autoclave quickly. After sealing the autoclave, the catalyst, MA, and FGE were further dried for 8 hours at 100 °C under a vacuum and then cooled to 15 °C. Subsequently, the autoclave was carefully purged with nitrogen, and nitrogen was alternately emptied and filled three times. Then 30 mL PO was injected into the autoclave with a syringe. The autoclave reactor was then pressurized to 5.0 MPa through a carbon dioxide cylinder. The copolymerization reaction was stirred at 60 °C for 40 hours, then the reactants were cooled to room temperature to release pressure. The obtained hard block product was dissolved in chloroform containing 5% hydrochloric acid solution to decompose the catalyst. The organic layer was washed to become neutral and slowly dripped into six times the volume of strongly stirred ethanol to precipitate the copolymer, which is called PPC-MF. The PPC-MF was alternately dissolved in chloroform and precipitated three times in ethanol to remove a small amount of propylene carbonate, then dried to constant weight at 80 °C in a vacuum, and the yield was calculated.

### 2.3. Characterization and Measurements

Fourier transform infrared spectroscopy (FT-IR) measurements were carried out on a Thermo Scientific Nicolet 6700 spectrometer equipped with attenuated total reflection (ATR) accessories.

^1^H NMR spectra were determined by Bruker DRX-400 spectrometer (Bruker Co., Rheinstetten, Germany) with chloroform-d as the solvent.

The average molecular weights of polymers were determined by gel permeation chromatography (GPC) system (Waters 515 HPLC pump, Waters 2414 detector) with tetrahydrofuran as an eluent. Polystyrene standards with a polydispersity of 1.02 were used to calibrate the GPC system.

The gel contents were determined by ASTM D2765 method. The sample was refluxed in boiled chloroform for 24 hours. The insoluble part was dried to constant weight at 80 °C in a vacuum. The gel content is defined as the weight percentage of the insoluble part in the sample. Data were recorded as the average of three parallel measurements.

Thermogravimetric analysis (TGA) was measured on a PerkinElmer simultaneous thermal analyzer (STA 6000). Samples were tested under 40 mL·min^-1^ nitrogen flow from 25–400 °C at a heating rate of 20 °C·min^-1^.

Differential scanning calorimetry (DSC) measurements were carried out under high-purity nitrogen flow in the temperature range of –25–200 °C, at a heating rate of 10 °C·min^-1^, using a Q100 TA instrument (New Castle, DE, USA). The onset of the change of heat capacity with temperature is regarded as the *T*_g_.

The hot-set test was carried out in an oven. The dumbbell-shaped specimen was loaded with 0.14 MPa and the reference length was marked as *L*_0_ (*L*_0_ = 20 mm). The load specimen was placed in an oven at 60 °C. After 15 minutes, the length between the markers was measured and recorded as *L*_1_. Then the load was released. After 5 minutes of relaxation at 60 °C, the specimen continued to relax to no longer shorten at room temperature. The length between the markers was measured and recorded as *L*_2_. The hot-set elongation and permanent deformation were calculated according to (*L*_1_ − *L*_0_)/*L*_0_ × 100% and (*L*_2_ − *L*_0_)/*L*_0_ × 100%, respectively.

The mechanical properties were tested at 23 °C using a CMT 6104 electronic tensile tester according to ASTM D368. The cross-head speed was 50 mm·min^−^^1^. The data were recorded as the average value of five parallel determinations. The dumbbell-shaped specimens for the tensile tests were prepared by hot-pressure molding followed by cutting by a dumbbell cutter.

## 3. Results and discussions

### 3.1. Synthesis

As expected, when PO/CO_2_/MA/FGE copolymerization was conducted, a unit linked to four CO_2_/PO copolymer arms forms once MA and FGE participate in the reaction, and random joining of these many units produces connected PPC chains (Scheme 1). The copolymerization results indicate that the gel really forms after introducing MA and FGE (Table 1). The gel contents increase from 11.0%–26.1% with the increase of MA and FGE from 1–4 mol % of PO, respectively, and the yields of copolymers roughly doubled. The presence of gel contents combined in the IR spectra indicates that MA and FGE are inserted into the backbone of PPC and the networks are successfully formed in one pot.

The FT-IR and ^1^H NMR measurements were used to characterize the structure of the copolymers. As shown in Appendix A, compared with PPC, the copolymer had new characteristic FT-IR absorption peaks at 1636 and 757 cm^-1^ that were assigned to the C=C stretching vibration and C–H out-of-plane bending vibration in the cis-disubstituted alkene of the bridged-ring units generated from the D-A reaction, respectively [43,44]. The other peaks at 2986, 1741, 1456, 1381, 1230, 1070, 977, and 788 cm^-1^ were similar to those of PPC and were ascribed to carbonyl and the open ring of PO [10,36]. It indicates that the MA and FGE monomers were incorporated into PPC successfully. The soluble fraction of copolymers had similar ^1^H NMR peaks with PPC (δ, ppm): 5.00 (s, CH), 4.20–4.27 (m, CH_2_), and 1.31 (s, CH_3_) (Appendix A). They were attributed to carbonate linkages formed by alternating CO_2_/PO copolymerization. In addition, there exist two peaks at 3.72, 3.57, and 1.18 ppm, which were assigned to CH, CH_2_, and CH_3_ from ether linkages, respectively [42,45]. The absence of signals of incorporated MA or FGE unit displays that they were almost confined to the gel which is not soluble in chloroform-d. The number-average molecular weights (*M*_n_s) of uncross-linked parts of the copolymers was slightly higher than that of PPC (Table 1). This further demonstrates that the uncross-linked parts of the copolymers contained almost no MA and FGE units. At the same time, there were no ^1^H NMR signals (C=CH) such as the peak at 7.05 ppm of MA [46] or 6.27 ppm of FGE (Appendix A). This demonstrates that the above mentioned FT-IR peaks at 1636 and 757 cm^-1^ were not from the monomer MA or FGE but instead the incorporated units generated from them.

### 3.2. Thermal Properties

The degradation temperatures of the PPC-MFs were significantly improved compared with that of PPC. As shown in Figure 1, Appendix A and Appendix A, the 5% weight-loss degradation temperature (*T*_d,__−5%_) of PPC-MFs was from 249.7–271.3 °C with various feed content of MA and FGE from 1–4 mol% of PO, whereas it is only 217.1 °C for PPC. Two maximum weight-loss degradation temperatures (*T*_d,max_) of 239.0 °C and 254.2 °C arose in the DTG curve of PPC (Appendix A). They generated from chain scission and unzipping reaction, respectively, based on the decomposition mechanism of PPC that was explored using thermogravimetric analysis/infrared spectrometry techniques [47]. Accordingly, the *T*_d,max_ of copolymers increased from 282.6–288.6 °C with an increase in MA and FGE. Combined with the fact that the gel content also increased gradually, the great improvement in thermostability was attributed to the formation of cross-links in PPC matrix because the cross-links obviously restricted the unzipping reaction. On the other hand, the *T*_g_s of the copolymers is lower than that of PPC after introducing MA and FGE. The former is 11.1–13.4 °C and the latter is 25.7 °C (Figure 2 and Appendix A)**.** The decrease of *T*_g_ can be explained from the following facts. First, in the case of CO_2_/other monomer copolymerization, the monomer MA or FGE had a negative influence on the *T*_g_ of the copolymer due to their flexible structure [10,48]. For example, the *T*_g_ of P((FGE-*co*-GME)C) copolymers was –25–2 °C with various FGE contents [40]. On the other hand, cross-linking modification of PPC reported previously has a slightly positive impact on *T*_g_ [13,14,30,35,36]. Moreover, the content of MA/FGE was low with less than 4 mol% of PO in this case, so the *T*_g_s were less than that of PPC but do not fall below 0 °C.

### 3.3. Dimensional Stability

It is well known that there is a severe cold-flow phenomenon for PPC because of both its low *T*_g_ and amorphous phase, so that it softens and deforms while being held in hand. Therefore, it is urgent to improve the dimensional stability of PPC. This performance was explored by hot-set elongation tests. As shown in Figure 3 and Appendix A, the hot-set elongations and permanent deformations of PPC-MFs reduced to 62.5% and 6.5%, respectively, with an increase in the feed contents of MA and FGE. In contrast, it was 310.7% and 157.2% for PPC, and the latter was 2318% more than that of PPC-MF-4. It was also observed that PPC-MF-1 had more hot-set elongation than PPC. The cross-links had a positive effect on dimensional stability while the low *T*_g_ had a negative effect. When the gel content was less, the low *T*_g_ played a major role, leading to the above phenomenon. It is clear that the permanent deformation of PPC-MF-4 reduced significantly as the gel content increased further. The improvement in dimensional stability was also observed intuitively by applying intense heat to the polymers, such as burning them (Video S1 and S2). Melting droplets emerged during PPC combustion while no droplets dropped for PPC-MF-4, even though its *T*_g_ was lower by almost 50% compared with that of PPC. These phenomena also prove that the networks formed in the copolymers and that these networks enabled the copolymers to have more dimensional stability at higher temperatures than PPC.

### 3.4. Mechanical Properties

The tensile results are listed in Table 2 and the stress–strain curves are seen in Appendix A. It was unexpected that the tensile strengths of copolymers decreased compared with that of PPC and the fracture strengths were greater than the yield strengths, unlike when rigid cross-linkable third monomers such as pyromellitic dianhydride or bicyclo[2,2,2]oct-7-ene-2,3,5,6-tetracarboxylic dianhydride were used to prepare cross-linked PPC that exhibits enhanced mechanical strengths than PPC [35,36]. The flexibility of MA and FGE incorporated into PPC chains significantly reduced the *T*_g_s of the copolymers (Figure 2 and Appendix A), which had a negative effect on the mechanical strength but a positive effect on the toughness. In this case, the low *T*_g_ played the dominant role, nevertheless the cross-linking helped to enhance mechanical strength. This indicates that the structures of the cross-linkable third monomers also had a significant effect on the mechanical strength of PPC, which did not necessarily increase after cross-linking.

## 4. Conclusions

We present a novel one-pot synthesis of PPC with cross-linked networks by introducing two cross-linkable monomers, MA and FGE, in the PO/CO_2_ copolymerization. The obtained copolymers are thermoplastic and displayed good elasticity and thermal and dimensional stability. They have potential applications as thermoplastic elastomers. Here the D–A reaction was selected as a way for constructing cross-links networks. We inferred that other reactions will also play the same role. That is, the two cross-linkable monomers contain reactive groups with each other and epoxy, carboxylic anhydride or lactone groups which can participate in the PO/CO_2_ copolymerization. Combined with our previous research on cross-linked PPC [35,36], the structures of cross-linkable monomers, like rigidity or flexibility, have an obvious effect on *T*_g_ and mechanical strength of copolymers. However, the cross-links always improve the thermal and dimensional stability whether the cross-linkable monomers are rigid or flexible.

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
