# Peer review of "A Novel One-Pot Synthesis of Poly(Propylene Carbonate) Containing Cross-Linked Networks by Copolymerization of Carbon Dioxide, Propylene Oxide, Maleic Anhydride, and Furfuryl Glycidyl Ether"

_polymers, 2019, doi:10.3390/polym11050881_

Round 1

Reviewer 1 Report

The article represents a novel work in the field of polymerization reaction in one-pot for the preparation of PPC polymers with cross-linked networks using CO2, PO, MA and FGE. I would like to denote the well structured and discussion of the results, facilitating reading to the reader.

Therefore, I recommend the work for its publication after some minor comments:

Lines 149-163: In FTIR and 1H NMR results, I suggest to support the assignation of peaks with additional references

Lines 178-187: Regarding thermal properties discussion, authors mention the decrease of Tg value as negative factor, but its true that a change in the properties of polymers can be specified as improvement or not depending on their final application, please specified why it implies a negative connotation.

Author Response

Response to the expert 1:

The article represents a novel work in the field of polymerization reaction in one-pot for the preparation of PPC polymers with cross-linked networks using CO2, PO, MA and FGE. I would like to denote the well structured and discussion of the results, facilitating reading to the reader.

Therefore, I recommend the work for its publication after some minor comments:

Point 1: Lines 149-163: In FTIR and 1H NMR results, I suggest to support the assignation of peaks with additional references

Respose 1: The references for assignment of FTIR and 1H NMR peaks have been added in revised manuscript. They are seen in Lines 201, 205,206 and 211.

Point 2: Lines 178-187: Regarding thermal properties discussion, authors mention the decrease of Tg value as negative factor, but its true that a change in the properties of polymers can be specified as improvement or not depending on their final application, please specified why it implies a negative connotation.

Respose 2: What we meant to say was that Tgs decease after introducing MA and FGE. The use of the word negative may give rise to ambiguity. In revised manuscript, the sentence that “the influence of introducing MA and FGE on the glass transition temperatures (Tgs) of the copolymers is negative compared with that of PPC.” has been changed to “the Tgs of the copolymers is lower than that of PPC after introducing MA and FGE.

Reviewer 2 Report

In this article the authors have carried out an extension of their previous work (Reference 50, Polymers, 2018, 10, 552). In both papers the authors have done very similar studies where in the present work they have used maleic anhydride (MA) and furfuryl glycidyl ether (FGE) for the synthesis of cross linked PPC. In the previous work they have used bicyclo(2,2,2)oct-7-ene-2,3,5,6-tetracarboxylic

dianhydride (BTCDA). The current study shows that the degradation temperature of PPC-MFs is improved with respect to PPC only. Dimensional Stability and permanent deformation was also studied. In my opinion the work is a good expansion of the previous work and can be published after following revisions.

1. Line 32. The term 'Novel' should be deleted from the abstract

2. Line 158, 159. The authors should write three NMR peaks instead of writing two peaks, which will explain the three different groups. Although the peaks at 3.57 and 3.71 ppm are very small they are well separated. I would suggest an increase in the concentration of the compound or if there is a problem with solubility then another NMR solvent can be used.

3. I would suggest to perform powder XRD for additional characterization of the polymer. 

Author Response

Point 1: In this article the authors have carried out an extension of their previous work (Reference 50, Polymers, 2018, 10, 552). In both papers the authors have done very similar studies where in the present work they have used maleic anhydride (MA) and furfuryl glycidyl ether (FGE) for the synthesis of cross linked PPC. In the previous work they have used bicyclo(2,2,2)oct-7-ene-2,3,5,6-tetracarboxylic dianhydride (BTCDA). The current study shows that the degradation temperature of PPC-MFs is improved with respect to PPC only. Dimensional Stability and permanent deformation was also studied. In my opinion the work is a good expansion of the previous work and can be published after following revisions.

Line 32. The term 'Novel' should be deleted from the abstract

Response 1: It has been deleted in revised manuscript.

Point 2: Line 158, 159. The authors should write three NMR peaks instead of writing two peaks, which will explain the three different groups. Although the peaks at 3.57 and 3.71 ppm are very small they are well separated. I would suggest an increase in the concentration of the compound or if there is a problem with solubility then another NMR solvent can be used.

Response 2: The expert is correct. The two signal peaks at 3.57 and 3.72 ppm are written separately in revised manuscript. They are assigned to CH2 and CH from ether linkages, respectively. Chloroform-d is a good solvent for the polymers (except for gels). The ether linkages contents (the repeating units of m in Figure S4) are much less than the carbonate linkages (the repeating units of n in Figure S4) in the polymers. That is, CO2 and PO are almost alternatively copolymerized. Continuous insertion of PO in the polymerization, which forms ether linkages, is rare. The main signal peaks of the carbonate linkages become very high when the concentration increases. Figure S4 has been revised.

Point 3: I would suggest to perform powder XRD for additional characterization of the polymer.

Response 3: PPC is amorphous. The crystallinity for PPC or modified PPC is a challenging topic. There are few reports about this subject. For example, the stereoregular PPC have been synthesized, but it still does not crystallize [Ref. 5 and 6 in manuscript]. In this work, the cross-linked networks generated by introducing MA and FGE and the resulting thermal and mechanical properties of the copolymers are the main investigations. The expert's suggestion of crystallinity characterization using XRD method is a good guidance. We will consider this aspect when we study suitable copolymerization systems in the future, for example, when a larger number of rigid third monomers are used in CO2/PO copolymerization.

Reviewer 3 Report

This study demonstrates the synthesis of poly(propylene carbonated)s (PPCs) with network structures crosslinked by the Diels-Alder (DA) reaction between the maleic anhydride and furfuryl glycidyl ether moieties. The one-pot polymerization of four monomers is probably an advantage of the strategy the authors adopted in this study. The synthesized crosslinked PPCs exhibited properties that differed from those of a linear PPC, which was suggested by a series of analysis such as DSC, TGA, a hot-set test, and a stress-strain test. The reviewer thinks that the manuscript may be acceptable for publication in Polymers after minor revisions.

1. The ratio of DA reactions

Is it possible to estimate the efficiency (or conversion) of the DA reaction? The gel content (insoluble content) is not so high (<27%), which suggests that the network structure is not so efficiently generated. Can the efficiency of the DA reaction and the gel content be improved by some methods, such as longer reaction time, a larger amount of catalyst, or higher CO2 pressure?

2. Monomer conversion

In relation to the above comment, it may be better to show the “monomer conversion values”, which is based on the charged amount of PO, FGE, and MA, of the products. In Table 1, the yields of products are listed; however, conversion values are more helpful to judge the efficiency of polymerization reactions.

3. The effect of Tg on the stress-strain test

As shown in Table 2 and Figure S6, a linear PPC exhibited a higher tensile stress than the crosslinked PPCs. The authors explained that the high Tg of a linear PPC (25.3 ºC) is responsible for the higher value. Indeed, the mechanical test was conducted at 23 ºC, a lower temperature than the Tg of a linear PPC. However, the reviewer think that the mechanical test should also be conducted at a temperature higher than the Tg for the purpose of confirming the effects of the crosslinks on the mechanical properties.

4. Illustration of a network structure shown in Scheme I

The illustration shown in Scheme 1 has a very uniform structure with controlled chain lengths between crosslinking points (it is like a tetra-PEG gel). However, the network structure generated in this study most likely has a heterogeneous structure with uncontrolled chain lengths between crosslinking points. Dangling chains are also probably formed. The illustration may mislead readers.

5. Molecular weights

Molecular weights of a linear PPC and soluble portions of crosslinked PPCs should be examined by some methods such as SEC.

Author Response

Point 1:

This study demonstrates the synthesis of poly(propylene carbonated)s (PPCs) with network structures crosslinked by the Diels-Alder (DA) reaction between the maleic anhydride and furfuryl glycidyl ether moieties. The one-pot polymerization of four monomers is probably an advantage of the strategy the authors adopted in this study. The synthesized crosslinked PPCs exhibited properties that differed from those of a linear PPC, which was suggested by a series of analysis such as DSC, TGA, a hot-set test, and a stress-strain test. The reviewer thinks that the manuscript may be acceptable for publication in Polymers after minor revisions.

 1 The ratio of DA reactions

Is it possible to estimate the efficiency (or conversion) of the DA reaction? The gel content (insoluble content) is not so high (<27%), which suggests that the network structure is not so efficiently generated. Can the efficiency of the DA reaction and the gel content be improved by some methods, such as longer reaction time, a larger amount of catalyst, or higher CO2 pressure?

Response 1: The contents of MA and FGE units incorporated into PPC chains may be estimated by the integral area of corresponding peaks like 6.27 ppm and 7.05 ppm (HC=CH) in 1H NMR spectra after the crosslinking bond is destroyed through DA inverse reaction. But these signal peaks are not detected once crosslinked (Figure S4). The gel may also contain incorporated MA and FGE units without DA reaction, which are not detected by 1H NMR. Therefore, it is not very accurate to estimate the degree of DA reaction by integral area of correlation peak of 1H NMR spectra.

The expert’s suggestion is very meaningful. It is possible to improve gel content by changing polymerization conditions. This research is a lot of work and needs further study. In this manuscript, we mainly report the one-pot method for preparing PPC with cross-linked networks by introducing different quantities of MA and FGE and its effect on properties. We thank the expert’s instructive suggestion and will consider the above aspect in future research.

Point 2: Monomer conversion

In relation to the above comment, it may be better to show the “monomer conversion values”, which is based on the charged amount of PO, FGE, and MA, of the products. In Table 1, the yields of products are listed; however, conversion values are more helpful to judge the efficiency of polymerization reactions.

Response 2: PO is used both feedstock and solvent in the CO2/PO copolymerization catalyzed by zinc glutarate described by the various research groups [1-4]. In this research field, the yields of polymers, or called as productivity, refer to activity of catalysts, and they are often provided in form of ‘gram of polymer per gram of cat., per gram of zinc or per mole of zinc’.

Point 3: The effect of Tg on the stress-strain test

As shown in Table 2 and Figure S6, a linear PPC exhibited a higher tensile stress than the crosslinked PPCs. The authors explained that the high Tg of a linear PPC (25.3 ºC) is responsible for the higher value. Indeed, the mechanical test was conducted at 23 ºC, a lower temperature than the Tg of a linear PPC. However, the reviewer thinks that the mechanical test should also be conducted at a temperature higher than the Tg for the purpose of confirming the effects of the crosslinks on the mechanical properties.

Response 3: The expert makes sense. The test temperature of 23 ºC is indeed higher than Tgs of the copolymers and lower than Tg of PPC. However, the tensile tests were conducted according to ASTM D638, in which the specified test temperature is 23 ºC. In fact, the positive effects of the crosslinks on the mechanical properties are seen from the stress–strain curves (Figure S6). Contrast to PPC, the fracture strength is much greater than the yield strength for the copolymers, which is the result by the crosslinks.

Point 4: Illustration of a network structure shown in Scheme I

The illustration shown in Scheme 1 has a very uniform structure with controlled chain lengths between crosslinking points (it is like a tetra-PEG gel). However, the network structure generated in this study most likely has a heterogeneous structure with uncontrolled chain lengths between crosslinking points. Dangling chains are also probably formed. The illustration may mislead readers.

Response 4: Our intention is to express the schematic diagram of the formation of the cross-linking structure, without considering the real situation, which would be misleading, and it has been changed in the revised manuscript according to the expert’s suggestion.

Point 5: Molecular weights

Molecular weights of a linear PPC and soluble portions of crosslinked PPCs should be examined by some methods such as SEC.

Response 5: The Molecular weights of a linear PPC and soluble portions of crosslinked PPCs have been measured by GPC and they are added in Table 1.

Attached References:

[1] Ree, M.; Bae, J.Y.; Jung, J.H.; Shin, T.J. A new copolymerization process leading to poly(propylene carbonate) with a highly enhanced yield from carbon dioxide and propylene oxide. J. Polym. Sci. Part A: Polym. Chem. 1999, 37, 1863–1876.

[2] Meng Y.Z.; Du L.C.; Tjong S.C.; Zhu Q.; Hay Allan S. Effects of the Structure and Morphology of Zinc Glutarate on the Fixation of Carbon Dioxide into Polymer. J. Polym. Sci. Part A: Polym. Chem.2002, 40, 579–3591.

[3] Marbach J.; Nornberg B.; Rahlf A.F. and Luinstra G.A. Zinc glutarate-mediated copolymerization of CO2 and PO – Parameter studies using design of experiments. Catal. Sci. Technol. 2017, 7, 28972905.

[4] Ang R.R.; Sin L.T.; Bee S.T.; Tee T.T.; Kadhum A.A.H.; Rahmat A.R.; Wasmi Bilal A. Determination of zinc glutarate complexes synthesis factors affecting production of propylene carbonate from carbon dioxide and propylene oxide. Chem. Eng. J. 2017, 327, 120–127.
